



# Feasibility of cell-phone camera Raman spectrometer for geological samples identification in field or mobile situations

Dinesh Dhankhar[1] and Matthew Wehner[2]

[1]Thermo Fisher Scientific, Hillsboro, Oregon
[2]Veratek Design, Escondido, California

**Correspondence:** Dinesh Dhankhar (dineshiist21@gmail.com), Matthew Wehner (nwgeologist@gmail.com)

**Abstract.** This note demonstrates the feasibility of integrating a Raman spectrometer in a cellphone (mobile phone) using its camera for identifying common minerals in different geologic materials. The cell phone modification is low cost ($\sim$ \$50 USD) and can be constructed by individuals, thus opening up the possibilities for "democratizing" Raman spectrometry in the geosciences and make it possible for it to be used in a wide range of field conditions (i.e., core logging, field mapping,
marketplace) and combined with other data types like GPS and optical photography for added value. The combination of Raman spectra with other data and the possibility to link it through Wi-Fi or cell data to a database opens new possibilities in terms of applications. It is imagined that with suitable apps or software, it could be used by anyone since the technology and spectral hardware exist to support such devices.

## 1   Introduction

Raman spectroscopy, a well-known and essentially non-destructive material analysis technique, is particularly suited for geologic applications (Griffith, 1969; Rathmell et al., 2021). Raman spectra represent the vibration energies of molecular bonds present in a sample, thereby providing a plethora of information about the constituents of a sample. A very common geologic application for Raman spectrometry include identification of minerals and other material phases in situ (Vaskova, 2011). Because of its sensitivity to molecular characteristics and its utility for solids, liquids, and even gas phases in situ, it has a place
as a common forensics tool in geoscience laboratory settings for characterizing materials. This has included using Raman spectroscopy to scan gems, for either confirming authenticity or identify unknown gems (Jenkins and Larsen, 2004).

In the geoscience disciplines, it has been used for heavy mineral studies, petroleum fluid characterization, thin-section petrography (Mao et al., 1987; Chung and Ku, 2000), etc. It is even hoped it can be used to detect evidence of life signs on Mars (e.g., Hutchison et al, 2014). For instance, on Mars there is a Raman spectrometer called Scanning Habitable Environments
with Raman & Luminescence for Organics & Chemicals (SHERLOC) that searches in ultraviolet for signs of life as part of NASA's Perseverance Mars mission (i.e., Bhartia et al., 2021).

In more earthly settings, specifically for organic-rich shales, Tuschel (2013) demonstrated that it is possible to quantify proportions of kerogen, calcite, and titanium oxide phases (can even distinguish between anatase and rutile, which are polymorphs). Raman spectroscopy has been explored as a tool for assessing the thermal maturity of organic matter in shales with





some promising results, while there is room for achieving a more robust methodology. (Sauerer et al., 2017; Hackley and Lünsdorf, 2018; Jubb et al., 2018; Lupoi et al., 2019).

The applications of Raman in micro-imaging are diverse and can be coupled with a wide range of complementary imaging instruments, like SEM, BSE, and EDS (i.e., Yuan et al., 2022). Obviously, the ability to combine Raman spectrometry with other methods has great advantages in terms of being able to use the molecular identification to confirm or disprove research

questions that require comparison with other data types.

While tabletop and lab-based Raman spectroscopy will remain important, it would be advantageous if a handheld Raman spectrometer was widely available at low cost, particularly if it was integrated with mobile phone devices which already have many other sensors that would enhance and simplify data collection of Raman spectrum with its metadata, for instance with GPS coordinates and optical photography, and this would enable more people to utilize Raman spectroscopy in everyday

situations and even education (Ozcan, 2014). The wide range of applications of portable Raman for applications beyond geoscience has been anticipated (Vandenabeele et al., 2014). There would greatly increase the ability of scientists and the laymen to screen interesting or unusual geologic samples onsite without the need to disturb or remove the samples, which would allow more selective screening for samples and in return reduce the amount of disturbance. Furthermore, the ability to take Raman spectra and connect it with data collected with existing geologic mobile apps would enhance its utility in sensitive

areas or remote locations. In addition, it is shown that cell-phone camera Raman spectrometers can be extremely cost effective, making it useful for widespread use as opposed to specialized compact instruments which cost 2-3 orders of magnitude more, lending itself to big data analytics if spectra collection could be collected from thousands of users and cross-correlated with spectra libraries. Several options currently exist for utilizing existing spectra libraries. They include: IRUG (described in Lomax et al., 2013) and RRUFF, the latter described by Culka and Jehlička (2019). These could serve as a starting point for spectra

identification and compositional "unmixing" of complex material for estimates of its constituent phases.

## 2    MATERIAL AND METHODS

Cell phone camera Raman spectrometers make use of the extremely good low light sensitivies of the modern cell phone cameras. In general, Raman scattering is an extremely weak effect with only about 1 in a million photons on average undergoing this effect. Raman scattering effect involves exchange of energies between the excitation light (which is usually a laser) and

the molecular bond vibrations. Vibrating molecular bonds can increase or decrease the energy of the scattered light by the amount equal to their vibration energies, thereby producing a specific energy shift in the scattered light which allows for precise identification of the molecular bonds in a material (i.e., McMillan, 1989).

It was shown that a cellphone camera Raman spectrometer, using right angle geometry between excitation and scattered light collection as shown in Fig 1 A could be constructed with optical components worth $\sim$ \$50 to record good quality Raman

spectra from chemical and biological molecules (see appendix of Dhankhar et al., 2021). This system works particularly well for liquid samples.





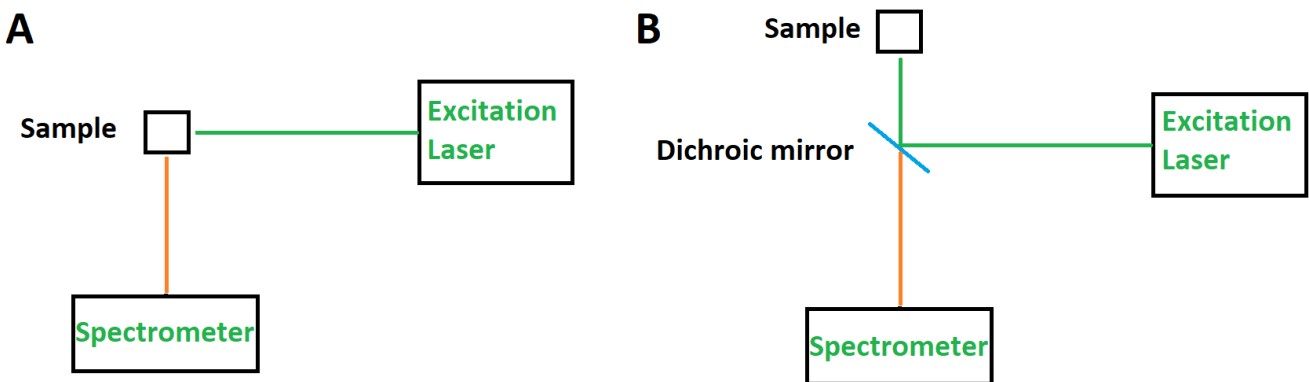

**Figure 1.** Two common geometries for excitation and collection of Raman spectra. Right angle geometry (A) and backscattered geometry (B).

Many geological samples, on the other hand, are solid, and for solid samples using the backscattered geometry of excitation and collection of Raman spectra is more convenient (See Fig. 1 B). Therefore, we constructed and recorded data from a system that collects the Raman spectra from solid samples using a cellphone camera as a detector in a backscattered geometry (Fig 1 B). The reason, solid samples benefit from backscattered geometry is partly, because, it is convenient to put the focused spot of the excitation laser on the solid surface to detect the photons that undergo Raman scattering. A disadvantage of this geometry is that it requires use of dichroic mirrors and/or filters to attenuate the strong excitation light scattering and reflecting back from the solid surface into the spectrometer. In our experiments, we utilized a dichroic mirror was obtained from Thorlabs (Part number DMLP550T).

In order to record the data, we utilized a 10x microscope objective lens and a dichroic mirror. The rock samples were placed at the focal distance of the microscope objective and excited by a 532 nm (50 mW) diode laser. The data recording and processing details are provided in (Dhankhar et al., 2021). Briefly, the spectra were recorded using a Google Pixel cell phone camera, in the night sight mode, and Proshot app. The raw images were recorded along with the jpg images. The resolution of the constructed system was $\sim 50 \text{ cm}^{-1}$. The resolution can be changed by changing the input slit size and/or the groove density of the diffraction grating used. The diffraction grating utilized in the constructed system was a low-cost transmission grating with groove density of 1000 grooves per millimeter. For more details on the design and construction of the spectroscope, please refer to (Dhankhar et al, Rev. of Sci. Ins. 2021). Wavelength calibration of cellphone camera Raman spectrometer was performed by using the Raman bands of a known sample of ethanol.

## 2.1 RESULTS

Some of the results obtained from the setup described are shown in Fig 2. We have included the Raman spectra of the same samples, recorded from a benchtop Raman spectrometer (model is Horiba Xplora) system for comparison. For both systems,





**Figure 2.** Results of Raman spectra obtained from cellphone camera Raman spectrometer of gypsum, calcite and diamond (D, E, and F, respectively). The recorded Raman spectra of the same samples by a laboratory benchtop spectrometer (Horiba Xplora) are shown on the left (A, B, and C for gypsum, calcite and diamond, respectively). For both the Raman spectrometers, excitation wavelength was 532 nm.

the excitation wavelength was 532 nm. The excitation laser power was 50 mW for the cellphone camera Raman spectrometer whereas it was 25 mW for the benchtop Raman spectrometer system.





### 2.1.1 DISCUSSION

80 Results show that cellphone camera Raman spectrometer had sufficient sensitivity to detect the characteristic Raman shifts for minerals like gypsum and calcite. There are several benefits of this approach, one, this is considered a non-destructive testing method (NDT) so useful for scenarios when the sample needs to be saved or used for another type of test. Two, it can more easily distinguish between silicates and carbonates, some of the most common mineral categories in nature. It can analyze minerals that are opaque in optical microscopy and amorphous materials (i.e., opal, a hydrated silica, which is essentially 85 invisible to XRD, a standard tool for mineral identification).

However, there are also several challenges associated with it. Fluorescence is a known issue because some materials can fluoresce strongly (orders of magnitude greater than the intensity of Raman bands) due to crystal defects or presence of certain elements (i.e., Mn or Fe). In some cases, the fluorescence spectra itself could be recorded and used for material/crystal identification (e.g., ruby, a corundum phase, has distinct wavelengths of fluorescence emission resulting from Cr) or even 90 identification of the type of impurities present in a known crystal. This could be an alternative use of the spectrometer described in this paper. It is also possible to add different color lasers in the setup to record both Raman and fluorescence spectra.

There are several ways to solve the problem of fluorescence spectra interfering with observation of Raman bands. The most obvious one is to change the wavelength of the excitation laser light. As the excitation shifts to longer wavelengths, the fluorescence typically decreases. For excitations in infrared (i.e., 780 nm) the fluorescence interference is typically very low. 95 Because most cameras have filters that remove the non-visible portion of the signal, it would probably be easier to include the lasers as part of a separate camera system on the cellphone with its own optimized filter. This is not necessarily unreasonable as it is already standard for cellphones to have more than one camera. Another way is to let the material sit under bright laser excitation for some time that quenches the fluorescence, making Raman bands more visible. In other words adjust the scan time for each Raman spectrum acquisition. Yet another way is to tune the laser wavelengths slightly either through heat or current, 100 and then take a difference of the spectra recorded under two conditions. This technique is known as Shifted Excitation Raman Difference Spectroscopy (SERDS) and is based on the idea that the fluorescence spectra remain unchanged with slight shifts is the excitation wavelengths, however the Raman lines move with changing the excitation wavelength. By taking a difference of the two spectra, the fluorescence signal cancels out leaving the Raman signal.

As mentioned earlier, in some cases, the fluorescence in could provide additional information for certain compounds like 105 fluorophores used in fluorescence spectroscopy (more relevant for organic chemistry). However, in the context of geoscience and mineralogy, there are certain minerals or impurities (known as activators) in minerals that fluoresce under UV excitation. Mineral samples that fluoresce under UV excitation are often highly sought by private individuals for their mineral collections. Certain minerals nearly always fluoresce like scheelite and autunite, while other minerals ordinarily not fluorescent under UV will fluoresce because of impurities that act as activators for fluorescing under UV light. Examples of activators include 110 Mn in calcite or dolomite (Robbins, 1987) but other known activators include REEs, Cr, Fe, Pb, and ions like $UO_4^{+2}$ or $S_2^{-2}$. Fluorescent minerals are grouped by whether they fluoresce under longwave UV (365 nm) or shortwave UV (254 nm).





Fluorescence can also be caused by more complicated processes that involve crystal lattice defects, which is relevant for covalently bonded or semiconductor minerals such as sphalerite or diamond (Modreski and Aumente-Modreski, 1996).

Also keep in mind that for this paper, no attempt was made to optimize or apply advance signal processing to the spectra, so it is expected that signal processing and probably even machine learning (Jahoda et al., 2021, describes using machine learning for mineral ID from Raman and other types of spectra) would improve the results presented in this paper if applied.

In the oil & gas industry, there are at least two cases where fluorescence may be detected while using a Raman spectrometer. The first is that geologic samples could contain organic material like petroleum, bitumen, and coal. The topic of fluorescence in organic matter is an entire topic unto itself so will not be covered in detail in this paper. Sometimes one encounters natural petroleum seeps or archived geologic samples that have oil stains (from a petroleum-containing reservoir), so a quick scan with a mobile-mounted Raman spectrometer could verify what one suspects. The other application known to the authors is in unconventional shale gas samples, sometimes volcanic ash beds are found (can be mm or meters thick) and these are known to fluoresce easily under UV excitation. A well-known example are the ash beds found in the Cretaceous Eagle Ford Shale of Texas, USA; studies indicate it has 55-110 volcanic ash beds and has been the topic/mentioned in multiple studies because of its impact on petroleum production (Amin et al., 2021; Lehrmann et al., 2019; Zeng et al., 2018; Xu et al., 2016; Lock, 2014) because it affects the fracture propagation that result from fracking and is easily detected/verified by fluorescence in drill core or outcrop.

It is not presumed that an integrated cellphone camera and Raman spectrometer would replace laboratory-based Raman spectrometers. But it would increase the number of users (whether the casual layperson, student, or even as a screening tool in more professional settings) and enable them to get this capability for comparatively little cost. Importantly it would have great value in the "exploration" stage where one needs to quickly identify samples for further analysis or confirm hunches like petroleum-saturated drill cuttings. It would have a "democratizing" effect and with appropriate educational exposure could stir many new studies and applications. The educational value is enormous as virtually anyone with a Raman spectrometer-equipped cellphone could use it on their personal device, especially if free or lost-cost apps existed to support the processing and identification of Rama spectra. This is in keeping with the general "democratizing" effect of including sensors in mobile phones that can be used by apps installed on said device for measurement and imaging as noted by Ozcan (2014). However, there are many possible scenarios where having a cellphone based integrated Raman spectrometer and optical camera would be advantageous.

One, it can be used for identifying minerals, for situations ranging from authentication at a gem show to in the field while hiking. In this case it is an informal check to confirm its ID or perhaps get a clue of something unusual or unexpected. Conceivably it could be used in an informal method in the mineral or petroleum exploration business. It is true that in the mineral or petroleum exploration business, it is already common to use portable XRF or handheld Vis-NIR for mineral identification for exploration projects for economic minerals or petroleum, so a cellphone camera Raman Spectrometer could have utility in core shacks or during field work.

Two, it could be used on gemstones. This is a reasonable application because gemstones are comparably more homogenous than typical rocks that are aggregates of multiple mineral phases. Also, the number of possibilities is considerably constrained



compared to the total number of known minerals (the International Mineralogical Association recognizes 5863 mineral species as of November 2022). Because one can encounter gemstone in casual settings (i.e., social settings or street markets), having a Raman spectrometer integrated with one's cellphone would make it convenient to do an immediate scan of gemstone of interest

and assess the likelihood of its authenticity on the spot without the delay and expense of a formal assay by a gemologist.

Three, we suppose that in the near future a Raman spectrometer could be a standard sensor included with the already proliferating number of standard sensors in cellphones. Cellphones that have an integrated Raman spectrometer could have the capability to collect Raman spectra nearly in real time for geologic samples in the field and for each spectrum record its GPS location, an optical photo, and any notes entered by user (whether by audio-to-text, touch-screen entry, or stylus selecting from

drop-down menus). Basically, it could greatly complement existing apps (i.e., ones that collect data like strike and dip with its GPS location) such as FieldMove Clino (Vaughan et al., 2014; Bubniak et al., 2020) or Rocklogger (Matsimbe, 2021).

Four, as noted earlier it may have utility for detecting/evaluating fluorescent minerals. This is an example where detection of fluorescence is a positive when it is usually a negative in other applications of Raman spectroscopy. While the use of a shortwave or longwave UV lamp is obviously preferred for prospecting and observing fluorescent minerals, the ability to

determine in a daytime field setting or in a gem show whether a mineral significantly fluoresces could have some utility.

Five, as already mentioned, there is in the oil & gas industry at least two use cases for the need to detect fluorescence. The first is that when dealing with drill cuttings from rotary drilling or similar destructive drilling methods, oil stains or droplets of petroleum can be detected by their fluorescence in addition to detection of organic molecular bonds by Raman spectrometry. This property of petroleum is already exploited by mudlogging companies when drilling new wells and typically is done with

UV light source and optical microscopy in a trailer lab type situation. However, this may be useful in situations where petroleum was not expected (i.e., a water or geothermal well). There is already a well-known instance of geothermal drilling encountering petroleum, the Romerberg Field in the Rhineland of Germany (Böcker et al., 2017). The ability to have Raman spectroscopy to immediately analyze the cuttings could provide additional information (i.e., natural petroleum or processed/partial processed petroleum typical in pipelines) to choose the correct response (i.e., calling environment emergency response). The second case

are volcanic ash beds, especially those found in organic-rich mudstones exploited as an unconventional source of petroleum, because those ash beds tend to fluoresce a bright yellow color under UV light. When it comes to studying drill cores, it is common to have a situation where one needs to examine core samples quickly or just a few hand specimens.

Six, this overlaps a bit with the application discussed in the previous paragraph (fifth scenario), environmental application is rather broad but could include in field soil assessment for various properties that have impact on engineering, environment,

or public safety. This could include tasks like correctly identifying spills as containing petroleum or other fluids. It may even have applications for monitoring water quality.

## 3   Conclusions

We do not pretend to have listed all the applications of Raman spectroscopy in geology but we note that the main theme in the potential applications of Raman spectroscopy is the ability to get nearly instantaneous results on geologic or fluid



composition that is important for making decisions where the timeframe of obtaining proper laboratory results or assays is much longer. In many industries, business decisions are made within minutes or days of field collection of samples and observations yet in some industries it can be weeks or months to get the assays. In this work, we obtained results from a cell phone modification that has already been demonstrated as being a low-cost option for post-manufacture modification which means virtually anyone worldwide can reproduce the set up and collect their own data. Furthermore, we allow for the possibility

that Raman spectrometer can become commonplace in cellphones as many other sensors have.

*Data availability.* All the data is available in the manuscript

*Author contributions.* D.D. designed the experiments and recorded the data. M.W. provided the samples for experiments. Both authors wrote the manuscript.

*Competing interests.* The authors declare that they have no conflict of interest.

*Acknowledgements.* Authors thank Professor Peter M. Rentzepis, Department of Electrical and Computer Engineering, Texas A&M University, College Station, Texas, USA, for generously allowing the use of his lab's benchtop Horiba Xplora Raman spectrometer for generating spectra for comparison of our results with the modified cellphone's Raman spectrometer.



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
