# Peer review of "Feasibility of cell-phone camera Raman spectrometer for geological samples identification in field or mobile situations"

_EGUsphere, 2023_

## Referee Comment (RC1)

**United States Department of the Interior**
**U.S. GEOLOGICAL SURVEY**
**Geology, Energy & Minerals Science Center**
**USGS National Center**
**12201 Sunrise Valley Drive, MS 954**
**Reston, VA 20192**

To: Oleg Korablev, Associate Editor, Geoscientific Instrumentation
From: Aaron Jubb
Date: October 10, 2023
RE: Reviewer comments for manuscript #egusphere-2023-2146

Title: Feasibility of cell-phone camera Raman spectrometer for geological samples identification in field or mobile situations

Authors: Dinesh Dhankhar and Matthew Wehner

This note describes the use of a cell-phone Raman spectrometer for geomaterials analysis. The major selling point of this approach is the democratization of access to a Raman spectrometer as the cell-phone modification is relatively inexpensive (<$500). While I found myself genuinely interested in this approach, the data presented is very limited and the discussion focuses more on hypothetical uses of the approach in place of a critical evaluation of the spectrometer's strengths and weaknesses for geomaterials analysis. The note's presentation is more in-line with a technical product note as one would find on a scientific instrument manufacturer's website instead of a research article. Major revisions, as detailed below, are needed before this is appropriate for publication.

Best,

Aaron M. Jubb, Ph.D.
Research Chemist
Geology, Energy & Minerals Science Center
United States Geological Survey
12201 Sunrise Valley Drive
Virginia 20192, USA

Recommendation: Major Revision

Major Revisions

1. Spectra are shown for hand samples of gypsum, calcite, and diamond. However, most geologic materials typically analyzed by Raman spectroscopy are spatially heterogeneous. How do the author's propose to measure samples where multiple phases are present in the probe spot? Or where one phase is the analytical target instead of adjacent phases? This article would benefit from the addition of data from geologically heterogeneous samples along with a discussion.

2. As I alluded to above, the discussion focuses on several potential applications (e.g., fluorescence) for the spectrometer without inclusion of any data. Inclusion of data toward this end will greatly bolster the discussion, which currently is speculative.

3. The cost estimates for the parts are overstated. For instance, the abstract states the cell phone modification is only ~$50, but as of 9/28/23 the dichroic mirror (Thorlabs part number DMLP550T) used was listed as $131.61. I suggest addition of a table with each part needed for the modification, the manufacturer, the price and the date purchased. This will provide context for readers interested in potentially attempting to make this modification to a cell-phone.

4. Provide actual citation to RRUFF database: Lafuente, B., Downs, R.T., Yang, H., Stone, N., 2015. The power of databases: the RRUFF project. In: Highlights in Mineralogical Crystallography, T. Armbruster and R. M. Danisi, eds. Berlin, Germany, W. De Gruyter, pp. 1-30.

5. Finally, more of an aside than a revision, but distinguishing between organic and mineral fluorescence, in my experience, is non-trivial. The authors are encouraged to carefully consider the discussion on Lines 117-127 and whether their cell-phone Raman spectrometer could accurately distinguish between organic and mineral fluorescence.

---

## Referee Comment (RC2)

To: Oleg Korablev, Associate Editor, Geoscientific Instrumentation

From: B Ramanan

Date: November 11, 2023

RE: Reviewer comments for manuscript #egusphere-2023-2146

Title: Feasibility of cell-phone camera Raman spectrometer for geological samples identification in field or mobile situations

Authors: Dinesh Dhankhar and Matthew Wehner

This note describes the feasiblity of using a cell-phone Raman spectrometer for geological applications. The major selling point of this approach is to simplify the Raman spectroscopy measurements under a variety of field conditions. The approach is quite interesting and has great potential in a wide range of applications. With minor revisions describing the SNR analysis with respect to various physical and environmental conditions, this note could potentially  make it for publication

Best,

B Ramanan,
Scientist
Laboratory of electro-optic sensors (LEOS)
Indian Space Research Organisation (ISRO)
Peenya, Bengaluru,
India

Recommendation: Minor Revision

Minor Revisions:

1. One of the major challenges in measuring Raman spectra of minerals is the positioning of the instrument for the efficient collection of Raman scattered photons. Is it possible for the authors to provide an analysis on the depth of focus of the system (i.e. SNR vs Working Distance) which could help the readers to understand the robustness as well as the flexibility of the system.

2. Since the instrument is intended to be used in real-world conditions, SNR analysis with regard to the environmental conditions, such as sunlight and dusty conditions. This information could help to understand the reliability of the system with respect to the varying environmental conditions.

---

## Editor Comment (EC1)

Dear Dinesh Dhankhar and Matthew Wehner,

Following the Reviewers comments I recommend a major revision of your manuscript in order to be accepted for publication in GI. Please provide a point-by-point answers to Reviewers' suggestions and the revised manuscript with marked differences.

I would also suggest a few general comments from myself:

1) Please pay appropriate attention to questions of the second Reviewer about collecting of photons and the SNR. In this manuscript you provide very limited technical information about the set-up, massively referring to your 2021 paper in RSI. However, there is no much information it that paper either. That makes difficult for an interested reader to reproduce your set-up.

2) In the 2021 RSI paper you discuss some problems of noise specific to cell-phone cameras. In the present paper you note a significant progress of phone cameras. From everyday life we know how different are cameras of different brands and models. A great share of their success is associated with digital post-processing of images. Maybe you wish to touch upon these issues when discussing the SNR.

3) At many instances you discuss a possible Raman cell-phone accessory or even a Raman built into a cell-phone. Yet there is a large technological gap between your low-cost prototype and these potential products. Their cost, including the needed application software and likely a specific cell-phone, might be much higher and it is not obvious how the potential cell-phone device competes with some hand-hold Raman device available on the market. Filling this tech gap is the most challenging while the applicability of a modest Raman device for mineral characterization hardly rises many objections. As the parameters of the set-up are still somewhat arbitrary, the value of technique validation is lowered. Such considerations might be behind the remark of the first Reviewer that the presentation resembles a product note. You might wish to change some accents accordingly.

With best wishes,
Oleg Korablev

---

## Author Comment (AC1)

We thank the reviewer for making many important comments and suggestions that makes the manuscript better. Below is our point by point response to the comments and the corresponding proposed changes in the revised manuscript.

1. Even though the data shown is only from pure isolated samples, one of the advantage of the Raman spectroscopy is that signal primarily comes from a very small localized spot of a few micrometer square area. Thus it is feasible to take Raman spectra of a heterogeneous sample by simply focusing the Raman excitation beam on different points of interest on the sample. In addition, it is also possible to place the sample on an xy(z) scanning stage and obtain a complete map of the materials present in an heterogeneous sample. Some work in this regard was done when we first reported cellphone camera based Raman spectrometer. Please see figure 11 in (Dhankhar et. al. Rev. sci. inst, 92(5), 2021). The discussion regarding the advantages and ways of obtaining Raman spectra of a heterogeneous sample will be added to the revised manuscript.

2. Fluorescence data of minerals will be added to the revised manuscript. Because fluorescence signals are typically orders of magnitude stronger than the Raman signals, it is possible to obtain very good quality fluorescence data.

3. Cost of ~$50 was estimated for a right angle geometry Raman spectrometer that does not require a dichroic mirror. A list of components for ~ $50 Raman spectrometer is provided in Table I of ( Rev. sci. inst, 92(5), 2021). The right angle geometry is quite suitable for the liquid samples, however for solid samples, a backscattered geometry gives stronger signal and more suitable for heterogeneous samples. An update component list and their corresponding costs and sources will be added to the revised manuscript.

4. The citation will be added to the revised manuscript.

5. There are several different ways of separating the fluorescence of different components when they are present together, such as by use of different excitation wavelengths (attaching different color lasers in this case). Discussion in the manuscript has been updated with examples and citations added.

With regard to Reviewer's comments that the manuscript's presentation is more like a product note than a research article, we would love to know specific reasons/places in the article that made reviewer reach this conclusion.

---

## Author Comment (AC2)

We thank the reviewer for bringing up the important subject of SNR and the depth of focus. Our point by point response to the reviewer comments are following:

1.  One of the major challenges in measuring Raman spectra of minerals is the positioning of the instrument for the efficient collection of Raman scattered photons. Is it possible for the authors to provide an analysis on the depth of focus of the system (i.e. SNR vs Working Distance) which could help the readers to understand the robustness as well as the flexibility of the system.

A. The optimum sample position with respect to the instrument is when the excitation laser is focused as perfectly as possible, on the sample, thereby generating the maximum amount of Raman signal. Raman signal generation is highly dependent on how tightly the excitation light is focused on the sample.

B. This means that if we utilize a focusing lens of higher NA (numerical aperture), we could obtain a stronger Raman signal because higher the NA , tighter (smaller) is the focused spot of the excitation light.

C. However, higher NA lenses are more difficult to focus because of their shorter working distances and very short depth of focus.

D. In our experiments, we utilized a 10x microscope objective lens that has NA of 0.25 and a 4x microscope objective lens that has NA of 0.1.

E. NA of 0.25 and 0.1 give us depth of field of approximately 8.5 and 53.2 micrometers, respectively. This could be computed using the approximate formula –

$$\text{Depth of field} \sim \lambda/N.A^2$$

Raman signal will start to reduce rapidly if the sample is placed more than the depth of field distance away from the optimum focus.

F. Since SNR value would be dependent on the strength of the Raman signal, it is fair to assume that for the best SNR, the sample must be placed within the depth of field distance of the optimum focal spot of the excitation light.

A few other things, that are worth mentioning with respect to the SNR discussion is that SNR not only depends on the efficient focusing, but also on efficient collection of the Raman scattered signals (one has to match the NA of the input optics with the spectrometer collection optics in order to collect the maximum signal). In addition, the Raman signals depend very significantly on the nature of sample as well as the excitation wavelengths. Some samples, such as diamond and single crystal silicon give a very strong Raman signal, whereas other samples may produce relatively weaker signals.

In general, the Raman signals depend inversely to the fourth power of the excitation wavelength; thereby shorter wavelength excitation lasers generate stronger signals. Although practically, the fluorescence generated by the shorter

wavelengths outweigh the benefits (unless excitation is in deep ultraviolet wavelength, shorter than 250 nm).

In addition, by using an excitation wavelength closed to electronic transitions in a sample, may results in resonance enhancement of the Raman signals, which is typically 1-2 orders of magnitude. For example, blue/green laser light is typically used to enhance the Raman signals from carotenoid-containing compounds.

Noise performance of the detector is obviously another important consideration. With modern cellphone camera sensors having much better hardware (typically with back thinned and back illuminated cmos sensor chips) as well as efficient software algorithms to eliminate noise, which are further becoming better everyday, make the scientific uses of them very promising.

SNR related above discussion will be added to the revised manuscript.

2. One of the major challenges in measuring Raman spectra of minerals is the positioning of the instrument for the efficient collection of Raman scattered photons. Is it possible for the authors to provide an analysis on the depth of focus of the system (i.e. SNR vs Working Distance) which could help the readers to understand the robustness as well as the flexibility of the system.

With regards to environmental conditions, it is very important to eliminate any sunlight reaching the spectrometer. Traditionally this is achieved by having a hollow black color tubing between sample and the excitation laser lens with a small opening at the laser focus. The sample can then be placed directly at the laser focus against this opening eliminating the sunlight entering the spectrometer. Additional shielding from an opaque cloth can also be utilized.

Although not applicable for cellphone Raman spectrometer; it is possible to build solar blind Raman spectrometer with excitation in the deep ultraviolet wavelengths.

With regard to the day to day environment dust, it is not likely to effect the instrument performance by any significant amount. Typical working distance between the focusing lens to sample is 1-3 cm and scattering by dust in typical day to day environment would not reduce the laser excitation intensity at the sample by a significant amount. This discussion with respect to the environmental conditions will be added to the revised manuscript.

Sincerely,
Dinesh Dhankhar, Matthew Wehner